# Transcriptome diversity is a systematic source of variation in RNA-sequencing data

**Pablo E. García-Nieto**⊚, **Ban Wang**⊚ iD ⊚, **Hunter B. Fraser**\*

Department of Biology, Stanford University, Stanford, California, United States of America

⊚ These authors contributed equally to this work.
\* hbfraser@stanford.edu

**Data Availability Statement:** The code is available at https://github.com/pablo-gar/transcriptome_diversity_paper.

**Funding:** This work was supported by NIH grant 2R01GM097171-09. PEGN was supported by a

## Abstract

RNA sequencing has been widely used as an essential tool to probe gene expression. While standard practices have been established to analyze RNA-seq data, it is still challenging to interpret and remove artifactual signals. Several biological and technical factors such as sex, age, batches, and sequencing technology have been found to bias these estimates. Probabilistic estimation of expression residuals (PEER), which infers broad variance components in gene expression measurements, has been used to account for some systematic effects, but it has remained challenging to interpret these PEER factors. Here we show that transcriptome diversity–a simple metric based on Shannon entropy–explains a large portion of variability in gene expression and is the strongest known factor encoded in PEER factors. We then show that transcriptome diversity has significant associations with multiple technical and biological variables across diverse organisms and datasets. In sum, transcriptome diversity provides a simple explanation for a major source of variation in both gene expression estimates and PEER covariates.

## Author summary

Although the cells in every individual organism have nearly identical DNA sequences, they differ substantially in their function—for instance, neurons are very different from muscle cells. This is in large part because different genes are transcribed from DNA into RNA, a key step in the process known as gene expression. The measurement of RNA levels is an important tool in studying biology, but is complicated by many potentially confounding factors. To account for this, computational methods can correct for unknown confounders, but these do not provide any information about what these confounders are. Here we show that transcriptome diversity–a simple metric based on Shannon entropy–explains a large portion of variability in both gene expression measurements as well as the confounding factors detected by a leading method. This prevalent factor provides a simple explanation for a primary source of variation in gene expression estimates.

Bio-X Bowes Graduate Student Fellowship. The funders had no role in study design, data collection and analysis, decision to publish, or preparation of the manuscript.

**Competing interests:** The authors have declared that no competing interests exist.

## Introduction

Gene expression is a fundamental process required by all life forms and its high-throughput quantification has been an active area of research for over 25 years [1]. A key step in this process is the transcription of DNA into RNA.

A myriad of methods have been developed over the past decades to assess RNA levels, including low-throughput techniques like RNA hybridization (e.g. northern blots, FISH) and Sanger sequencing, as well as high-throughput methods like DNA microarrays and next-generation RNA-sequencing (RNA-seq). Each of these methods presents a unique set of advantages and technical difficulties. The main advantage of RNA-seq is its ability to measure expression levels of all non-repetitive genes in the genome, resulting in its widespread adoption for biological research [2]. Due to its simplicity and commercialization, researchers can readily prepare RNA and send it to sequencing centers, obtaining data in a matter of hours to a few days.

Even though there are multiple experimental methods to generate bulk RNA-seq data, it is now considered to be a standard practice, with most of them generating raw data in the form of short sequencing reads [2]. Similarly, while there are multiple computational tools to transform these sequencing reads into gene expression values, they generally follow these standard steps [3]: (1) performing quality control on the experiment and individual reads, (2) mapping reads to a reference genome to identify their gene-of-origin, (3) creating gene counts, and (4) transforming those counts into gene expression values to be compared across genes and/or experiments. This last step has proven to be non-trivial because gene counts in RNA-seq are of relative nature by design [4], i.e. the number of reads that are sequenced is many orders of magnitude smaller than the number of RNA transcripts in a cell population [5]. Thus, the read count of a gene depends on the counts of all other genes.

Computational methods have been developed to normalize and/or transform raw read counts to account for undesired effects caused by the relative nature of RNA-seq [4]. While the Transcripts Per Million (TPM) normalization has been used extensively, it has been shown to be problematic when there are major disparities in gene expression levels or sequencing depth across experiments [3]. The two most widely adopted methods that attempt to overcome issues of TPM are the "Trimmed Median of Means" (TMM) [6] and the "Median of Ratios" [7]. Despite some differences between the two, they both rely on creating a shared pseudo-reference expression vector (1 x # genes) from an expression matrix (# samples x # genes), and this vector is then used as a normalization factor across all samples. Since their conception, TMM and Median of Ratios have been extensively used for differential gene expression analysis and eQTL discovery, and they have been incorporated into the best practices of large consortia like the use of TMM in the Genotype-Tissue expression project (GTEx) [8].

Many factors can globally affect gene expression estimates. These include extremely highly expressed genes, sequencing depth differences among samples, ancestry, sex, age, sequencing technology, and RNA integrity [3,9–11]. In a heterogenous sample collection, not controlling for these effects can cause spurious results in downstream analysis.

In addition, there are other unmeasured and unknown systematic effects influencing gene expression estimates that need to be corrected prior to expression analyses [12]. This has been supported by the inference of broad variance components of gene expression matrices by probabilistic estimation of expression residuals (PEER) [13]. PEER can find one-dimensional arbitrarily scaled "hidden" factors that as a whole explain much of the global variation in gene expression across multiple samples. It has become a common practice for gene expression quantitative trait locus (eQTL) studies of heterogenous samples (e.g. GTEx) to use PEER hidden factors as covariates in models for eQTL discovery. This pipeline increases the sensitivity

**Fig 1. Illustration of transcriptome diversity.** Transcriptome diversity ($H_s$) was computed per sample based on Shannon entropy. $G$ is the total number of expressed genes and $p_i$ is the probability of observing a transcript for gene $i$. An example of two samples with three genes is shown, where one sample has a higher transcriptome diversity value ($H_1$) with more evenly distributed sequencing reads aligned to genes than the other sample ($H_2$) with one gene responsible for the majority of the sequencing reads.

of eQTL mapping [8], but has remained challenging to interpret because PEER factors *a priori* are not strongly associated with any known biological or technical source.

In this manuscript we show that Shannon entropy–a simple metric that assesses the transcriptome diversity of an RNA-seq sample [14]–explains much of the global variability in gene expression; is a major factor that PEER identifies; and is linked to a myriad of technical and biological variables. Shannon entropy was first developed as part of information theory to measure the level of "surprise" in a random variable [15], and has since been adopted in many different fields. In the biological sciences, Shannon entropy has been used in ecological studies to measure species diversity of a population, and it has been applied to gene expression to assess transcriptomic diversity [14]. Entropy is highly correlated with within-sample gene expression variance (Fig A in S1 File). For instance, the least diverse (and highest variance) transcriptome would have all transcripts from only one gene and the most diverse (lowest variance) transcriptome would express an equal number of transcripts across all genes (Fig 1). In other words, diversity reflects our ability to predict what gene a randomly chosen RNA-seq read comes from—less diverse transcriptomes are dominated by highly expressed genes, and are therefore more predictable.

## Results

### Transcriptome diversity explains a large portion of the variability in global gene expression estimates of RNA-seq samples

While analyzing published *D. melanogaster* RNA-seq data [16], we observed that transcriptome diversity as measured by Shannon entropy [14] (Fig 1; see Materials and methods: Transcriptome diversity calculation (Shannon entropy)) strongly correlates with the expression of many genes (measured by TMM). An example of one gene is shown in Fig 2A (Spearman's $\rho$ = 0.56, n = 851 flies, p-value = $3.3 \times 10^{-70}$).

To determine the extent to which transcriptome diversity predicts genome-wide gene expression levels, we further analyzed this *D. melanogaster* dataset [16]. TPM values have been shown to not properly account for sequencing depth differences across samples and for the influence of highly expressed genes on the rest of genes (usually referred to as RNA composition) [4]. TMM (and the similar Median of Means) is a more effective normalization method that in theory accounts for those effects and has been widely adopted for comparing gene expression across samples. Therefore, we used TMM as our major estimates to investigate the variation that transcriptome diversity is involved, and we performed analysis on TPM estimates in parallel (results shown in Figs H, I and J in S1 File). We observed that most TMM-

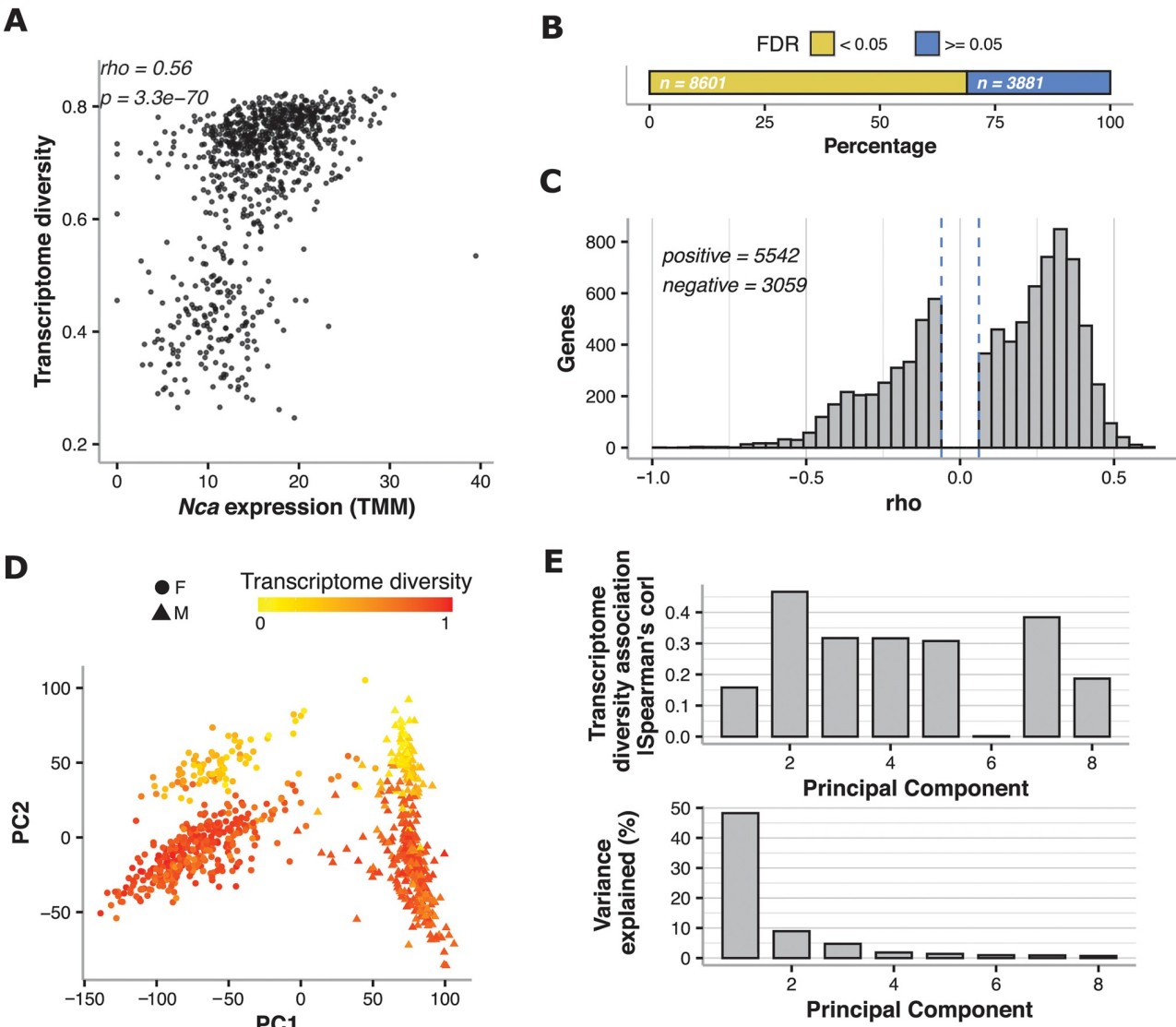

**Fig 2. Transcriptome diversity is associated with global TMM gene expression in D. melanogaster. A** Example of a strong association between the TMM expression of a gene encoding for a calcium-binding protein (*Nca*) and transcriptome diversity across samples from a large RNA-seq study [16]. **B** Percentage of genes whose expression was significantly associated with transcriptome diversity (as in A; BH-FDR < 0.05 in yellow) vs those that were not (BH-FDR > = 0.05 in blue). The actual number of genes is shown with white text. **C** Most of the significant associations between TMM estimates and transcriptome diversity are positive as shown here by the distribution of Spearman's correlation coefficients. Thresholds of Spearman's correlation coefficient corresponding to FDR < 0.05 are shown as dashed vertical lines. **D** Loadings from the first two principal components (PCs) from a principal component analysis done on the full TMM expression matrix; samples are colored by transcriptome diversity and the point shape corresponds to sex. **E** Absolute Spearman's correlation coefficients between transcriptome diversity and loadings of the first 8 PCs (top), and variance explained by each of those PCs of the full expression matrix.

based expression values were correlated with transcriptome diversity (68.9% of genes at Benjamini-Hochberg False Discovery Rate (BH-FDR) <0.05; Fig 2B; see Materials and methods: Expression associations with transcriptome diversity). Interestingly, most (64%) of the significant correlations are positive (Fig 2C; i.e. higher expression in samples with higher transcriptome diversity).

In agreement with the single gene correlations, a principal component analysis (PCA) shows that a substantial fraction of gene expression variation across samples can be explained

by their transcriptome diversity (Fig 2D and 2E; see Materials and methods: PCA analysis). PC1 mainly separates flies based on sex (which is known to affect expression levels in *D. melanogaster* [17]), but to a certain extent it is also correlated with transcriptome diversity (Fig 2D and 2E; Spearman's $\rho$ = -0.16, p-value = $3.7 \times 10^{-6}$), and PC2 is even more highly correlated with diversity (Fig 2D and 2E; Spearman's $\rho$ = -0.47, p-value< $2.2 \times 10^{-16}$). Overall, transcriptome diversity explains 4% of the TMM variance (S1 Table).

In sum, these results show that gene expression variation across RNA-seq samples can be partially explained by variation in transcriptome diversity across those samples.

## PEER "hidden" covariates encode for transcriptome diversity

Probabilistic estimation of expression residuals (PEER) is a method that was developed to extract a set of variables that explain maximal variability in a gene expression matrix with heterogenous samples [13]. These PEER factors may represent unmeasured global technical or biological information which can be used as covariates to reduce their confounding effects. While controlling for PEER covariates has proven to be useful in studies performing eQTL scans or differential expression analyses, the sources of these factors are unknown.

Based on the results of the previous section, we reasoned that PEER factors could partially encode for transcriptome diversity. To test this hypothesis, we computed 60 PEER factors (as recommended for this sample size [8]; see Materials and methods: Uniform processing) across the 851 flies in this dataset and performed correlation analysis with the corresponding transcriptome variability values of each sample. Out of the 60 PEER factors, 28 showed a significant correlation with transcriptome diversity (BH-FDR<0.05, Fig 3A left panel). Those 28 PEER covariates explain a substantially higher variance of the gene expression matrix (Fig 3A right panel; see Materials and methods: Variance explained of gene expression matrices), compared to the rest of the covariates which only explained a small fraction of the variance (Fig 3A right panel; see Materials and methods: Variance explained of gene expression matrices).

We devised a strategy to assess how much of the gene expression variance explained by PEER is accounted for by transcriptome diversity (see Materials and methods: Gene expression variance explained by PEER accounted by transcriptome diversity and other covariates). Results show that transcriptome diversity accounts for 9.2% of the variance explained by PEER in TMM expression (Fig 3B; difference of median $r^2$ values).

In sum these results suggest that a component of PEER "hidden" factors originates from differences in transcriptome diversity across samples.

## Transcriptome diversity explains a large portion of the variability of gene expression in GTEx

Given the extent to which transcriptome diversity explains gene expression variability in *D. melanogaster*, we wondered whether this might hold for other RNA-seq datasets as well. We thus decided to analyze GTEx data as this is one of the largest RNA-seq projects to date [8]. These data provide an excellent platform to perform comparisons across multiple individuals and tissues as it provides RNA-seq data for more than 17,000 samples across 948 human donors and 54 tissues.

We calculated transcriptome diversity values across samples in GTEx (excluding tissues with low number of samples, see Materials and methods: Uniform processing) and then assessed how much of the within-tissue variation in the expression matrices could be explained by transcriptome diversity. We first took the PCA approach described above for *D. melanogaster* data. PCs based on TMM expression estimates showed strong associations with transcriptome diversity (Fig 4A; 28 out of 49 tissues have the strongest correlation with PC1, median

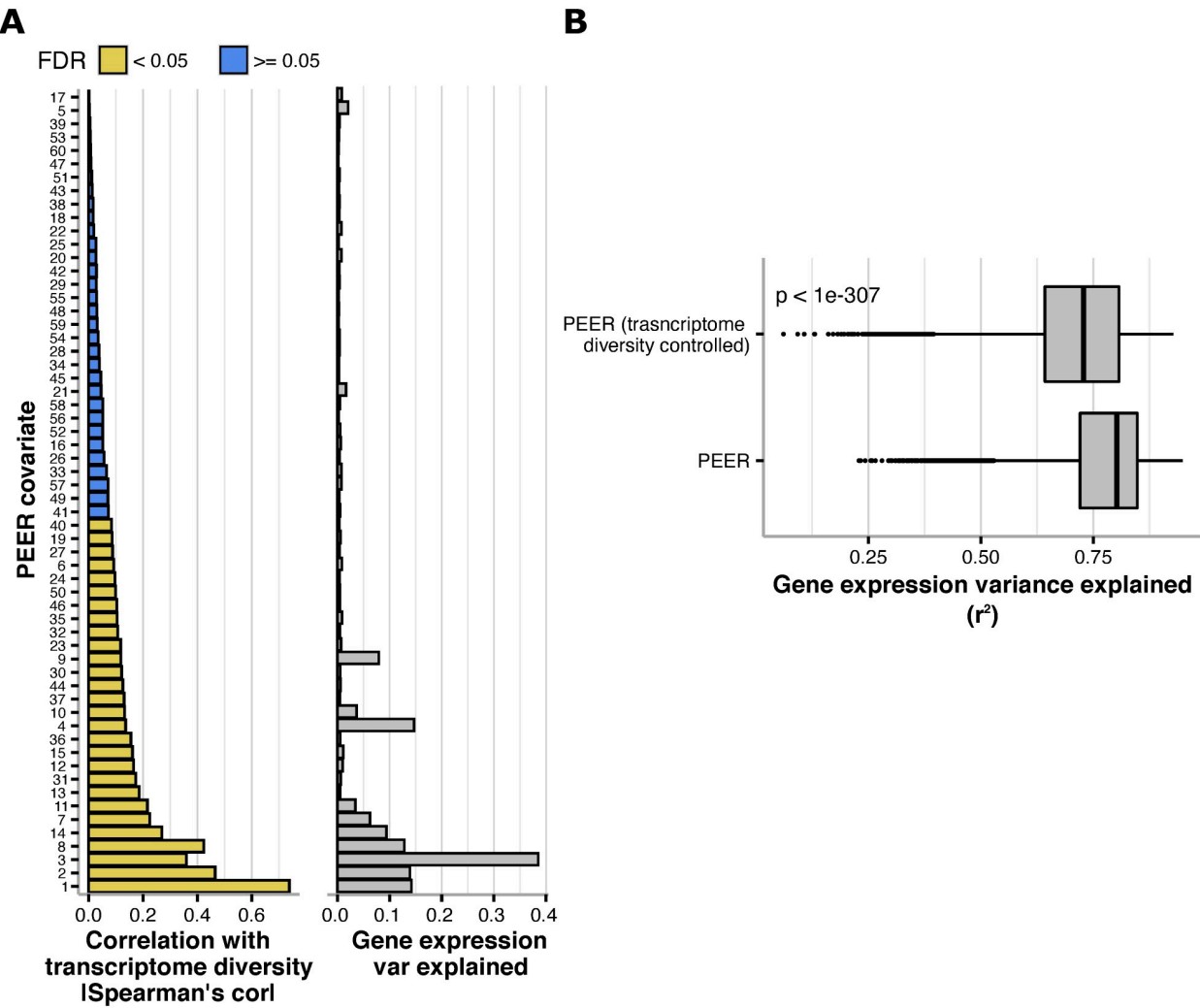

**Fig 3. Transcriptome diversity is a major factor encoded in PEER covariates. A** Left: Spearman correlation coefficients between transcriptome diversity values and the values of all PEER covariates obtained from the full expression matrix (see Materials and methods: Uniform processing) and colored by significance of correlation using BH-FDR. Right: Variance explained by each PEER covariate of the full expression matrix (calculated using a PCA-based method; see Materials and methods: Variance explained of gene expression matrices). **B** Boxplots showing the distribution of variance explained values ($r^2$) from linear regressions done on the expression of each gene using intact PEER covariates, or the residuals of regressions performed on the same PEER covariates using transcriptome diversity (transcriptome diversity controlled, see Materials and methods: Gene expression variance explained by PEER accounted by transcriptome diversity and other covariates). The p-values from a Mann-Whitney test are shown.

Spearman's $|\rho|$ = 0.56; 10 out of 49 tissues have the strongest correlation with PC2, median Spearman's $|\rho|$ = 0.48).

In line with our PCA results, most genes across all tissues in GTEx showed a significant correlation (BH-FDR<0.05) between TMM expression estimates and transcriptome diversity (median 63.7% of genes per tissue; Fig 4B), and as seen in *D. melanogaster* more of the significant correlations were positive (median 55.1% positive correlations; Fig 4B). To determine what attributes of a gene are predictive of the association between its expression level and transcriptome diversity, we performed multiple regression with gene length, GC content, and average gene expression level as independent variables (see Materials and methods: Factors contributing to dependencies of gene expression on transcriptome diversity). We found that

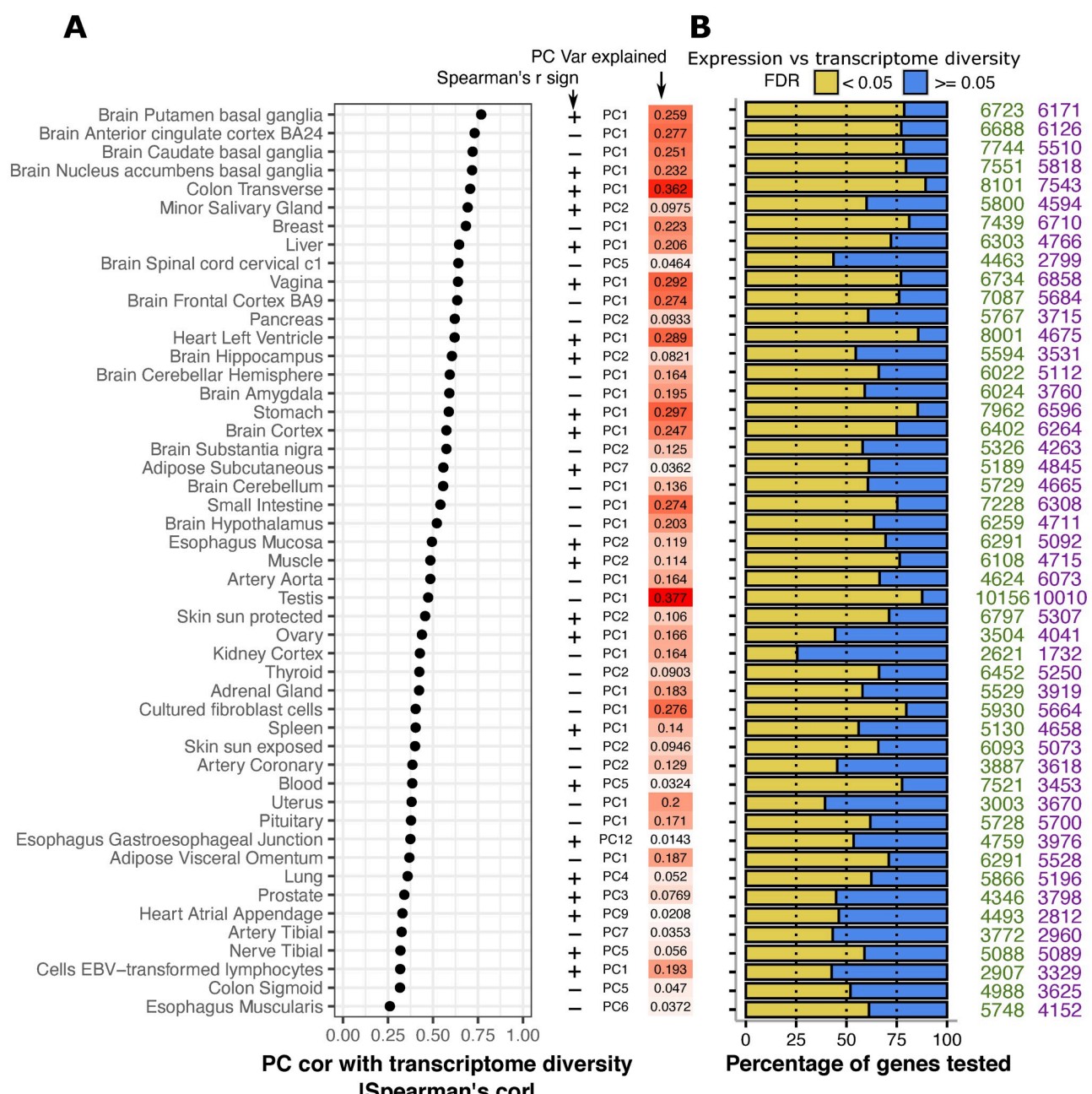

**Fig 4. Transcriptome diversity is associated with the expression of most genes across human tissues. A** For each GTEx tissue, the dot plot shows the absolute Spearman correlation coefficient between transcriptome diversity values and the loadings of a PC from a PCA performed on the full TMM expression matrix. To the right, the directionality of the correlation is shown (+/-) along with the PC used and its total variance explained. The PC with the highest correlation with transcriptome diversity is shown. Tissues are ordered by absolute Spearman correlation coefficient, and there is no pattern observed when ordered by sample size (Fig B in S1 File). **B** For each tissue, the percentage of genes whose expression TMM was significantly associated with transcriptome diversity (as in A; BH-FDR < 0.05 in yellow) vs those that were not (BH-FDR > = 0.05 in blue), the numbers on the right represent the directionality of the significant correlations (green are positive significant associations, and purple are negative significant associations). Significance was assessed using a linear regression approach (see Materials and methods: Expression associations with transcriptome diversity).

these three factors all contribute to the association of gene expression with transcriptome diversity. Longer and less highly expressed genes tend to have stronger associations between gene expression and transcriptome diversity in most tissues. GC content was the most consistent predictor of a gene's association between gene expression and transcriptome diversity (Fig C in S1 File), though the direction of this relationship varied across tissues.

Overall, transcriptome diversity explains a significant proportion of the global gene expression variance across GTEx tissues (median 6% TMM variance; S1 Table; see Materials and methods: Variance explained of gene expression matrices), with some exceptional cases like the brain putamen basal ganglia (15% TMM variance) and heart left ventricle (11% TMM variance).

Similar to the *D. melanogaster* data, the PEER covariates that explained the most GTEx gene expression variance were significantly correlated with transcriptome diversity (Fig D in S1 File). As an example, Fig 5A (top) shows that 20 out of 60 PEER covariates from blood were significantly correlated with transcriptome diversity and those covariates explained high levels of the gene expression variability (Fig 5A bottom; for all tissues see Fig D in S1 File).

To assess how much of the gene expression variance explained by PEER is accounted for by transcriptome diversity, we performed gene-based associations between expression and intact PEER factors, as well as PEER factors where transcriptome diversity was regressed out (identically to our *D. melanogaster* analysis, see Materials and methods: Gene expression variance explained by PEER accounted by transcriptome diversity and other covariates). Since GTEx data is heterogeneous we repeated this analysis regressing other metadata including biological and technical factors out of PEER factors (sex, sequencing depth, sequencing platform, PCR amplification method, and the first 5 PCs from the genotype matrix).

We found that among all variables we tested, transcriptome diversity accounted for the most variance explained by PEER factors from TMM expression values (see Materials and methods: Gene expression variance explained by PEER accounted by transcriptome diversity and other covariates). For example, in blood, transcriptome diversity accounted for 6.8% of the variance explained by PEER, in muscle 6.3%, and in sun-exposed skin 4.8% (Fig 5B, see S2 Table for all tissues). Although sequencing depth seemed as a strong factor and accounted for 6.1% of the variance explained by PEER in whole blood, this is only found in blood and not in other tissues, and transcriptome diversity is the strongest factor among all tissues that accounted for the largest variance explained by PEER (S2 Table). To our knowledge this is the first example of a known source of variability explaining PEER covariates to such an extent.

Overall, these results suggest that transcriptome diversity explains a significant amount of gene expression variance in RNA-seq data from diverse species and is also a major component of PEER covariates.

## Transcriptome diversity is associated with a variety of technical and biological factors

As an attempt to understand better the interconnection between transcriptome diversity and other features from RNA-seq we made use of published datasets (see Materials and methods: Data sources and data retrieval) that were designed to probe the relationship between technical and biological influences on gene expression estimates.

Among all variables tested for associations with transcriptome diversity we observed that sequencing depth was consistently and positively correlated with transcriptome diversity (Fig 6A and 6B). This association was not a simple consequence of read depth affecting transcriptome diversity, as random sampling of reads to equalize sequencing depth across samples did not change the relative transcriptome diversity values of samples. Interestingly, we

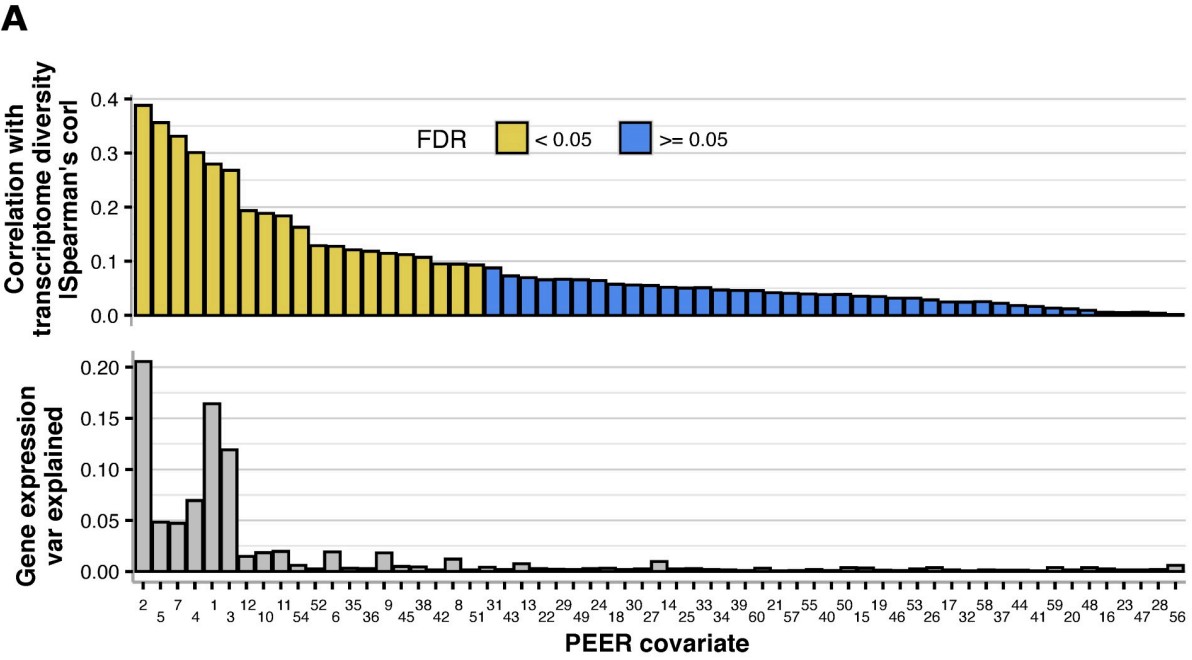

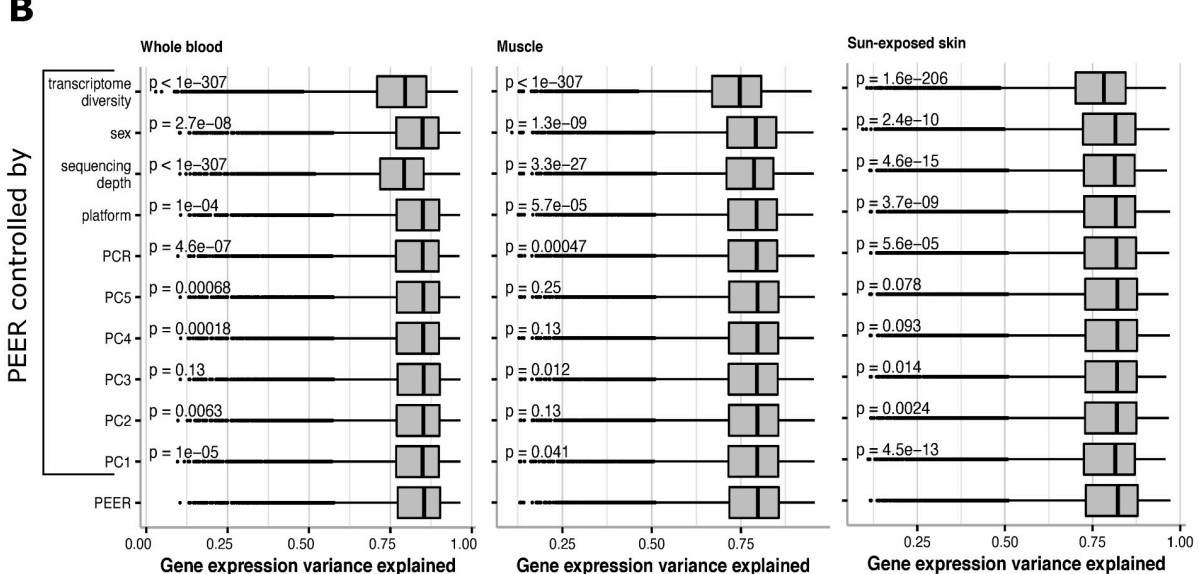

**Fig 5. In GTEx, PEER covariates correlate with transcriptome diversity. A** Top: Spearman correlation coefficients between transcriptome diversity values and the values of all PEER covariates obtained from the full GTEx blood expression matrix (see Materials and methods: Uniform processing) and colored by significance of correlation using BH-FDR. Bottom: Variance explained by each PEER covariate of the GTEx Blood full expression matrix (calculated using a PCA-based method; see Materials and methods: Variance explained of gene expression matrices). For all other GTEx tissues, see Fig D in S1 File. **B** Boxplots for three selected tissues whole blood, muscle, and sun-exposed skin, showing the distribution of variance explained values ($r^2$) from linear regressions done on the expression of each gene using either intact PEER covariates, or the residuals of regressions performed on the same PEER covariates using the variables shown (controlled PEER rows, see Materials and methods: Gene expression variance explained by PEER accounted by transcriptome diversity and other covariates). Mann-Whitney tests against the intact PEER covariates were performed for each of the controlled PEER distributions and the corresponding p-values are shown.

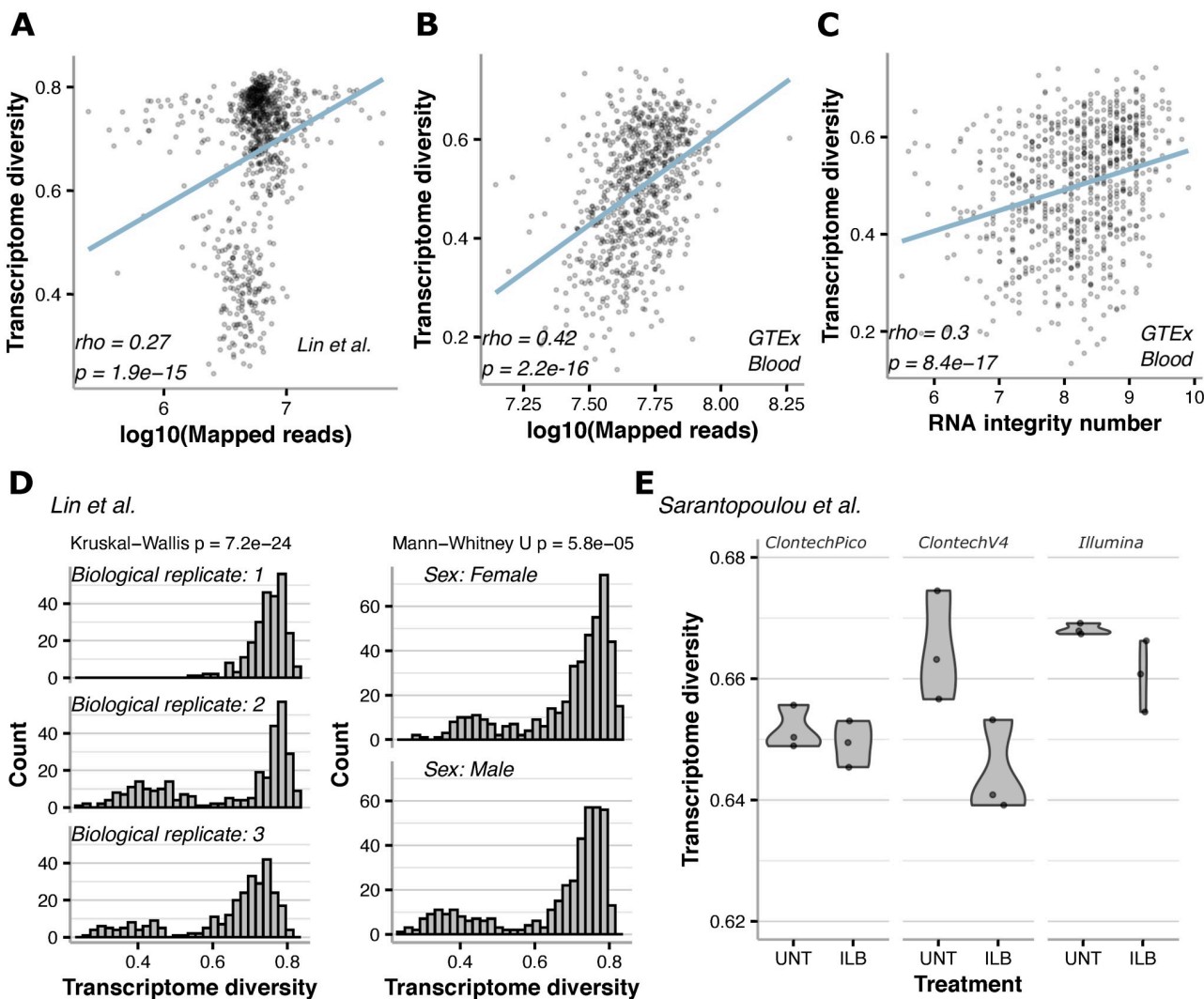

**Fig 6. Other technical and biological factors are associated with transcriptome diversity. A,B** transcriptome diversity was consistently associated with RNA-seq sequencing depth, shown for *D. melanogaster* [16] and GTEx blood. **C** RNA integrity also exhibited significant correlations with transcriptome diversity as shown here for GTEx blood. **D** RNA-seq data from *D. melanogaster* [16] show that transcriptome diversity can differ across biological replicates (left) as well as sex (right). **E** Different sequencing library preparations and perturbations result in varying transcriptome diversity distributions as shown by these violin plots. UNT is untreated mouse liver samples and ILB is Interleukin 1 beta treatment.

also found that RNA integrity was strongly associated with transcriptome diversity, with more fragmented RNA having an overall lower transcriptome diversity (Fig 6C). This suggests that factors making some samples more "sequenceable" (such as RNA integrity–as a consequence of sample condition and preparation) may affect both read depth and transcriptome diversity.

We then asked whether biological replicates lead to differences in transcriptome diversity. Lin et al. (2016) [16] performed 3 biological replicates of 17 *D. melanogaster* strains; within each biological replicate, they also included technical replicates and a mixture of both males and females were included for each strain. Variation in transcriptome diversity values were observed among technical replicates (Fig E in S1 File). Comparing transcriptome diversity distributions across biological replicates revealed significant differences (Fig 6D). While there was a significant difference in transcriptome diversity values when comparing male vs female

flies, the magnitude of this difference was smaller compared to differences among biological replicates (Fig 6D).

We then compared different RNA-seq library preparation methods by analyzing data from a study of mouse liver that compared three methods: Illumina TruSeq stranded mRNA Sample Preparation kit (Illumina), Takara Bio SMART-Seq v4 Ultra Low Input RNA kit (Clontech-V4), and Takara Bio SMARTer Stranded Total RNA-Seq Kit v2 –Pico Input Mammalian (Clontech-Pico) [18]. We observed clear differences between these three methods, as Illumina produced the highest transcriptome diversity values, followed by Clontech-V4, and then Clontech-Pico (Fig 6E). The original study examined differential expression after Interleukin 1 beta (ILB) treatment, and interestingly we observed that all ILB-treated samples had overall lower transcriptome diversity values (Fig 6E).

We also examined the variation of transcriptome diversity values among tissues in GTEx data. We found that some tissues had a much wider distribution of transcriptome diversity values than others (Fig F in S1 File). For example, samples from blood and all 13 sampled brain regions had substantial variation of transcriptome diversity values, suggesting that the effects of controlling for transcriptome diversity may be most pronounced in these tissues.

Finally, we asked whether different processing pipelines could affect transcriptome diversity. Arora et al. (2020) [19] compiled data from different sources that re-processed GTEx data from raw sequencing reads to gene counts (GTEx v6 [20], Xena from UCSC [21], Recount2 from John Hopkins [22], mskcc from cBio and mskccBatch from cBio [23]). These pipelines differ in quality-control filters, mapping procedures and counting techniques. We observed consistent and clear differences of transcriptome diversity values among these pipelines (Fig G in S1 File).

Altogether these results show that transcriptome diversity has complex associations with biological and technical aspects of RNA-seq, both from the experimental and computational sides.

## Discussion

Despite much research over the last decade, it has proven difficult to provide appropriate normalization methods to estimate gene expression from RNA-seq read counts. At the core of the problem lays a major limitation of RNA-seq: the number of sequenced reads is typically less than 0.01% of the total number of transcripts in a sample [5]. As a result, RNA-seq expression levels are relative quantities (referred to as compositional data [4]) where an increase in one gene's expression leads to a decrease in the relative expression of all other genes. As a consequence, comparing the expression of any given gene across different samples becomes a non-trivial issue, and multiple normalization methods have been developed to account for the relative nature of sequencing data. Ultimately, any differences (even minor) between any two samples in the distribution of reads across genes may cause global systematic changes in gene expression estimates.

Transcripts per million (TPM) was one of the first widely adopted normalizations, but it has been shown to be heavily affected by unusually highly expressed genes and sequencing depth differences [3]. While TMM [6] and the Median of Ratios [7] addressed some of these issues, they were not designed to account for overall sample differences in the distribution of reads across genes. In this manuscript we provide evidence that these differences result in pervasive effects on gene expression estimates that confound gene expression analysis.

To investigate the source of these confounding effects, we have used a metric that captures the distribution of reads across genes in an RNA-seq sample–transcriptome diversity based on Shannon entropy. Shannon entropy was first formulated to measure the level of surprise of a

random variable, and when applied to read counts it represents the diversity of the transcriptome. The transcriptome diversity value we used ranges from 0 (a sample with all reads mapping to one gene) to 1 (a sample with reads equally distributed across all genes; see Materials and methods: Transcriptome diversity calculation (Shannon entropy)). Throughout this study we showed that in a collection of samples, the expression of a gene across samples strongly correlates with transcriptome diversity and while correlations were prevalent in TMM estimates (Figs 2 and 4), and even more pervasive with TPM estimates (Figs H, I and J in S1 File). Moreover, these associations held for the vast majority of genes and across datasets that spanned different organisms (Figs 2, 4 and 6), and tissues (Fig 4). Overall, our results show that current normalization methods fail to account for the systematic effects captured by transcriptome diversity differences among samples.

Systematic effects on gene expression had been previously shown to exist. The PEER method was designed to produce a set of vectors that captures these effects from a multi-sample expression matrix [13]. We reasoned that since transcriptome diversity was capturing large portions of systematic effects on global gene expression, then PEER covariates could be capturing this information. Indeed, a substantial portion of PEER covariates significantly correlates with transcriptome diversity, and those covariates with the strongest correlations also explain the highest levels of gene expression variance (Figs 3 and 5; Fig D in S1 File). As a result, a significant fraction of the gene expression variance explained by PEER can be accounted for by transcriptome diversity (Figs 3B and 5B; S2 Table). Thus, a major factor that PEER is capturing can be encoded by this simple metric–transcriptome diversity.

However, transcriptome diversity can encode true biological signal itself, and not necessarily purely data bias. For example. We saw higher variation of transcriptome diversity for samples from all brain regions (Fig F in S1 File) and correspondingly, brain regions showed the strongest correlation between gene expression and transcriptome diversity (Fig 4A). The higher variation in transcriptome diversity from brain regions could reflect variation in cell type abundances across samples.

It is worth noting that all RNA data analyzed in this study originates from bulk sequencing where reads do not include a unique molecular identifier (UMI). The nature of this data makes it prone to be influenced by experimental artifacts during library preparation, for example PCR amplification leading to inaccurate transcript estimates. It is possible that transcriptome diversity is influenced by such artifacts. Applying the analyses presented in this study to single-cell RNA-seq data with UMIs would shed light on the influence of library prep artifacts on transcriptome diversity.

Here, we are not aiming to claim that transcriptome diversity should be used in place of PEER covariates or be the ultimate solution to normalize gene expression. Instead our goal is to bring researchers' attention to transcriptome diversity, a prevalent factor that could be a simple explanation for a major source of variation in gene expression studies. While PEER remains a powerful approach for correcting unknown confounding factors, a deeper knowledge of the sources of PEER factors—including transcriptome diversity—could lead to more precise and interpretable normalization approaches that avoid overcorrection [24], as well as improved experimental practices that minimize confounding.

## Materials and methods

### Ethics statement

Human data in this study was obtained from public GTEx repositories (https://gtexportal.org). Its privacy and ethical documentation can be found at: https://gtexportal.org/home/documentation. No personal identifiable information was used in this study.

### Data sources and data retrieval

For reproducibility a snakemake pipeline is provided at this study's github repo (https://github.com/pablo-gar/transcriptome_diversity_paper). This pipeline was used to download data from the original sources (see below).

The original data were processed uniformly to produce standardized matrices of read counts, TPM and TMM estimates. PEER covariates were calculated for some of them as mentioned in the main manuscript. Uniformly processed expressing matrices can be found in the S1 Text, the original data can be found in the following links:

- **Mouse data (Lin et al**. (2016) [16]). Raw count matrices and metadata were downloaded from GEO (https://www.ncbi.nlm.nih.gov/geo/query/acc.cgi?acc=GSE60314)

- **GTEx data**. Raw counts, TPM estimates, eQTL-ready expression matrices, and metadata were downloaded from GTEx v8 web portal (https://gtexportal.org)

- **GTEx data from different processing pipelines (Arora et al**. (2020) [19]). These data were compiled by Arora et al. (2020) [19] and made available at https://s3-us-west-2.amazonaws.com/fh-pi-holland-e/OriginalTCGAGTExData/index.html

- **Mouse data (Sarantopoulou et al**. (2019) [18]) Raw count matrices and metadata were downloaded from GEO (https://www.ncbi.nlm.nih.gov/geo/query/acc.cgi?acc=GSE124167)

### Transcriptome diversity calculation (Shannon entropy)

Shannon entropy was introduced to RNA-seq elsewhere [14]. Shannon entropy is defined as:

$$H = -\sum_i p_i log_2(p_i)$$

Where $i$ is an element (e.g. gene), and $p_i$ is the probability of observing element $i$. Thus, we define Shannon entropy as follows for an RNA-seq sample:

$$H = -\sum_i^G p_i log_2(p_i)$$

$$p_i = \frac{c_i}{l_i} \cdot \frac{1}{\sum_j^G \frac{c_j}{l_j}}$$

Where $G$ is the total number of expressed genes in a sample, $c_i$ and $l_i$ are the number of reads and effective length in base pairs of gene $i$, respectively. In words, $p_i$ is the probability of observing a transcript for gene $i$ in the RNA-seq library, which is equal to the number of reads mapping to that gene normalized by its effective length, and further divided by the sum of those values across all genes in the library. Using TPM instead counts, $p_i$ is defined as:

$$p_i = \frac{TPM_i}{\sum_j^G TPM_j}$$

We provide proof for the following in S1 Note.

$$H = -\sum_i^G p_i log_2(p_i) = -\sum_i^G \frac{TPM_i}{\sum_j^G TPM_j} log_2\left(\frac{TPM_i}{\sum_j^G TPM_j}\right)$$

$H$ ranges from *0 to log$_2$(G)*, where $H$ = 0 when all transcripts are from only one gene and $H = log_2(G)$ when an equal number of transcripts are measured across all genes. Therefore, we define the following to be able to compare transcriptome diversity across samples:

$$H_s = \frac{-\sum_i^G \frac{TPM_i}{\sum_j^G TPM_j} log_2\left(\frac{TPM_i}{\sum_j^G TPM_j}\right)}{log_2(G)}$$

$H_s$ ranges from 0 to 1 and this is the value that we refer as transcriptome diversity throughout the paper.

## Uniform processing

Raw read count matrices were downloaded from public sources (see above), except for Arora et al. (2020) [19] data. These count matrices were reformatted for uniform processing and the following calculations were done on them:

- TPM: Transcripts per million were calculated by adapting functions from the R package "scuttle" [25]. Effective gene lengths were defined as the cumulative length of exons from the mapping transcript.

- TMM: Trimmed median of means was calculated per each dataset using edgeR's functions *calcNormFactors* and *cpm* [26].

- Transcriptome diversity: Transcriptome diversity values were calculated per sample as described in section 'Transcriptome diversity calculation (Shannon entropy)'. See S3 Table for all values.

- PEER covariates: As recommended [8], for each dataset we first filtered out lowly expressed genes (only keeping those genes with at least TMM = 1 in 20% of samples). We then calculated *N* PEER covariates from the TMM matrix using the "peer" R package (https://github.com/PMBio/peer). Following GTEx guidelines [8] *N* was dependent on the number of samples of the dataset, *N* = 15 for up to 150 samples, *N* = 30 for 151–250 samples, *N* = 45 for 251–350 samples, and *N* = 60 for more than 350 samples. PEER covariates for GTEx samples were directly downloaded from the GTEx portal (https://gtexportal.org). See S4 Table for all values.

    The code used for all of these calculations is available at this study's github repo (https://github.com/pablo-gar/transcriptome_diversity_paper).

## Expression associations with transcriptome diversity

To test for the association between transcriptome diversity and the expression of individual genes, we used linear regressions with the following model:

$$y = \beta_0 + \gamma x + \epsilon$$

Where *y* is the expression of a gene across samples and *x* is the transcriptome diversity of the

corresponding samples. Significance is measured based on the p-value of a t-test performed on γ. P-values are adjusted using Benjamini-Hochberg False Discovery Rate.

To follow GTEx standards of association analyses, in all cases both *y* and *x* were normalized by converting the values into quantiles and mapping them to the corresponding values of the standard normal distribution quantiles.

## Factors contributing to dependencies of gene expression on transcriptome diversity

Gene length and GC content were retrieved via the R package "EDASeq" [27] using ENSEMBL gene IDs from Biomart database.

To test whether gene length, GC content or mean expression level can predict the association between gene expression vs. transcriptome diversity, we used multiple regression with the following model:

$$y = \beta_0 + \gamma_1 x_1 + \gamma_2 x_2 + \gamma_3 x_3 + \epsilon$$

Where *y* is the association level of the expression of a gene and transcriptome diversity, and $x_1$, $x_2$ and $x_3$ are the gene length, GC content and mean expression level (TPM or TMM) of the corresponding genes respectively. Significance is measured based on the p-value of a t-test performed on $\gamma_1$, $\gamma_2$ and $\gamma_3$. P-values are adjusted using Benjamini-Hochberg False Discovery Rate.

## PCA analysis

PCA was performed for each dataset matrix, both using TPM and TMM expression estimates. The R function *prcomp* was used with default parameters.

## Variance explained of gene expression matrices

Throughout the study for certain datasets, we aimed to calculate how much variance of gene expression can be explained by transcriptome diversity and PEER covariates.

To accomplish this, for a given dataset, we first performed PCA as described above. This allowed us to reduce the dimensionality of the expression matrix as well as know how much variance each of the PCs explains. For example, we computed $v_i$ as the variance explained by $PC_i$ of the expression matrix. For each of the query factors we calculated Pearson correlations between all PCs and one vector (e.g. transcriptome diversity). $r_i^2$ (the square of the Pearson correlation coefficient) is computed as the coefficient of determination between the query factor (e.g. transcriptome diversity) and $PC_i$.

Multiplying the $r_i^2$ from one of these correlations (e.g. transcriptome diversity vs PC1) by the variance explained by that PC provides a partial variance explained by the query vector, and adding these values across all correlations from that query vector (e.g. transcriptome diversity) provides the total variance explained by the query vector of the gene expression matrix.

$$v_{total} = \sum_{i=1}^{n} r_i^2 \times v_i$$

Where $v_{total}$ is the total variance and *n* is the total number of PCs. The query factor can be PEER factors to compute the total variance explained by PEER factors of the gene expression matrix.

To account for the normality assumption of Pearson correlation, they were calculated on rankit-normalized vectors, i.e. mapping values to a standard normal distribution based on quantiles.

## Gene expression variance explained by PEER accounted by transcriptome diversity and other covariates

We first performed associations between PEER covariates and gene expression. $r_1^2$ (the square of the Pearson correlation coefficient) is computed as the coefficient of determination between intact PEER and gene expression.

We then regressed out transcriptome diversity from PEER factors using a linear regression and used the residuals to repeat the gene expression associations. $r_2^2$ is computed as the coefficient of determination between "regressed out" PEER and gene expression.

The difference in $r^2$ values between the intact PEER and gene expression association test ($r_1^2$) vs the "regressed out" PEER factor residuals and gene expression association test ($r_2^2$) divided by the variance explained by PEER (see section 'Variance explained of gene expression matrices') represents the amount of variance explained by PEER that can be accounted for by transcriptome diversity.

When computing for gene expression variance explained by PEER accounted by other covariates such as sex, platform, PC1 etc, the same method was applied as for transcriptome diversity, and only replacing transcriptome diversity with other covariate (e.g. sex) when regressing out from PEER factors.

## Supporting information

**S1 File.** Supplementary figures (Figs A–J). **Fig A in S1 File. Transcriptome diversity is highly correlated with within-sample gene expression variance in both TMM and TPM estimates**. Transcriptome diversity across samples from a large RNA-seq study in *D. melanogaster* [16] shows significant associations with gene expression variance both in TMM estimates (left) and TPM (right). Variance was computed using TMM and TPM values respectively. Spearman correlation coefficients and p-values were computed and shown in each panel. **Fig B in S1 File. Transcriptome diversity is associated with the PCs across human tissues related to Fig 4A**. For each GTEx tissue, the dot plot shows the absolute Spearman correlation coefficient between transcriptome diversity values and the loadings of a PC from a PCA performed on the full TMM expression matrix. To the right, the directionality of the correlation is shown (+/-) along with the PC used and its total variance explained. The PC with the highest correlation with transcriptome diversity is shown. Tissues are ordered by sample size. **Fig C in S1 File. Gene length, GC content and gene expression level are associated with the correlation of gene expression with transcriptome diversity**. For each GTEx tissue, -10*log10(p-value) was computed from a multiple regression of association level of gene expression to transcriptome diversity on gene length, GC content and gene expression level. For visualization purpose, 1e-50 was added to all p-values. The black dashed line shows the cut-off p-value equal to 0.05. **a** In TMM estimates, lower average gene expression tended to have stronger association between gene expression and transcriptome diversity across all tissues. GC content showed negative correlation in most tissues, i.e. lower GC content has stronger association, except positive correlation observed in 7 tissues (adipose subcutaneous, artery aorta, brain cortex, breast mammary tissue, pancreas, thyroid and uterus). Gene length showed significant association in 42 out of 49 tissues (except adipose visceral omentum, adrenal gland, brain amygdala, liver, skin not sun exposed suprapubic, skin sun exposed lower leg and thyroid), and longer genes showed stronger association in most tissues (except artery tibial, brain amygdala, skin not sun

exposed suprapubic and testis). **b** In TPM estimates, longer genes and lower average gene expression tended to have stronger association between gene expression and transcriptome diversity across all tissues except that gene length showed negative correlation in testis. GC content showed significant association in 44 out of 49 tissues (except adipose subcutaneous, brain cortex, breast mammary tissue, pancreas and uterus), and most correlations between GC content and diversity association are negative except artery aorta, brain cortex, breast mammary tissue and thyroid. **Fig D in S1 File. PEER covariates associated with transcriptome diversity explain a large fraction of variance in global gene expression**. For each tissue, the Spearman correlation coefficient between transcriptome diversity values and the values of all PEER covariates were computed and colored by significance of correlation using BH-FDR (BH-FDR $> = 0.05$ in blue, BH-FDR $< 0.05$ in yellow). The variance of the full expression matrix explained by each PEER covariate was computed and projected on the y axis. **Fig E in S1 File. Variation observed in transcriptome diversity among technical replicates**. 117 flies with 2 technical replicates were tested by Lin et al. (2016) [16]. For each pair of technical replicates, samples with higher transcriptome diversity values were assigned into technical replicate 1 and samples with lower transcriptome diversity were assigned into technical replicate 2. Spearman correlation coefficient and p-value were computed and shown. **Fig F in S1 File. Large variation observed in transcriptome diversity across tissues in GTEx**. Transcriptome diversity values' distribution are shown in violin plots for all tissues in GTEx, indicating a wide range of variation for transcriptome diversity among tissues. **Fig G in S1 File. Differences on RNA-seq computational pipelines have a strong impact on transcriptome diversity**. Five computation pipelines for RNA-seq data (mskcc, mskccBatch, recount2, v6 and xena) are shown to have impacts on transcriptome diversity across tissues (data from Arora et al. (2020) [19]). Kruskal-Wallis rank sum tests were performed, and p-values are shown in each panel. 14 out of 15 tissues (all except salivary) showed significant differences in the distributions of transcriptome diversity values among the five pipelines. **Fig H in S1 File. Transcriptome diversity is associated with global gene expression in D. melanogaster, similar analysis on TPM estimates related to** Fig 2. **a** Example of a strong association between the TPM expression of a gene (Nca) and transcriptome diversity across samples from a large RNAseq study [16]. **b** Percentage of genes whose expression was significantly associated with transcriptome diversity (as in a; BH-FDR $< 0.05$ in yellow) vs those that were not (BH-FDR $> = 0.05$ in blue). The actual number of genes is shown with white text. **c** Most significant associations using TPM estimates are positive, as shown here by the distribution of Spearman's correlation coefficients (rho) between transcriptome diversity and gene expression. **d** Loadings from the first two principal components (PCs) from a principal component analysis done on the full TPM expression matrix; samples are colored by transcriptome diversity and the point shape corresponds to sex. **e** Absolute Spearman's correlation coefficients between transcriptome diversity and loadings of the first 8 PCs (top), and variance explained by each of those PCs of the full expression matrix. **Fig I in S1 File. Transcriptome diversity is associated with the expression of most genes across human tissues, similar analysis on TPM estimates related to** Fig 4. **a** For each GTEx tissue, the dot plot shows the absolute Spearman correlation coefficient between transcriptome diversity values and the loadings of a PC from a PCA performed on the full TPM expression matrix. To the right, the directionality of the correlation is shown (+/-) along with the PC used and its total variance explained. The PC with the highest correlation with transcriptome diversity is shown. **b** For each tissue, the percentage of genes whose expression TPM was significantly associated with transcriptome diversity (as in a; BH-FDR $< 0.05$ in yellow) vs those that were not (BH-FDR $> = 0.05$ in blue), the numbers on the right represent the directionality of the significant correlations (green are positive significant associations, and purple are negative significant associations). Significance was assessed using a linear regression

approach (see Materials and methods). **Fig J in S1 File. In GTEx PEER covariates correlate with transcriptome diversity on TPM estimates related to** Fig 5B. Identical to Fig 5B but for whole blood, muscle and sun-exposed skin GTEx samples in TPM estimates. Boxplots showing the distribution of variance explained values ($r^2$) from linear regressions done on the expression of each gene using either intact PEER covariates, or the residuals of regressions performed on the same PEER covariates using the variables shown (controlled PEER rows). Mann-Whitney tests against the intact PEER covariates were performed for each of the controlled PEER distributions and the corresponding p-values are shown.
(PDF)

**S1 Table. Gene expression variance explained by transcriptome diversity for all datasets analyzed in this study.**
(TSV)

**S2 Table. Median variance explained values ($r^2$) from linear regressions done on the GTEx expression of each gene using either intact PEER covariates, or the residuals of regressions performed on the same PEER covariates using the indicated variables.**
(TSV)

**S3 Table. Transcriptome diversity of all samples analyzed in this study.**
(TSV)

**S4 Table. PEER covariates.**
(TSV)

**S1 Note. Mathematical proof for transcriptome diversity equation from TPM values.**
(PDF)

**S1 Text. Link to google drive file with all expression matrices used in this study.**
(TXT)

## Acknowledgments

We would like to thank members of the Fraser Lab for helpful feedback.

## Author Contributions

**Conceptualization:** Pablo E. García-Nieto, Hunter B. Fraser.

**Data curation:** Pablo E. García-Nieto, Ban Wang.

**Formal analysis:** Pablo E. García-Nieto, Ban Wang.

**Funding acquisition:** Hunter B. Fraser.

**Investigation:** Pablo E. García-Nieto, Ban Wang.

**Methodology:** Pablo E. García-Nieto, Ban Wang.

**Software:** Pablo E. García-Nieto, Ban Wang.

**Supervision:** Hunter B. Fraser.

**Visualization:** Pablo E. García-Nieto, Ban Wang.

**Writing – original draft:** Pablo E. García-Nieto, Ban Wang, Hunter B. Fraser.

**Writing – review & editing:** Pablo E. García-Nieto, Ban Wang, Hunter B. Fraser.

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
