## [Decision Letter · Decision Letter 0]

16 Nov 2021

Dear Dr. Fraser,

Thank you very much for submitting your manuscript "Transcriptome diversity is a systematic source of bias in RNA-sequencing data" for consideration at PLOS Computational Biology.

As with all papers reviewed by the journal, your manuscript was reviewed by members of the editorial board and by several independent reviewers. In light of the reviews (below this email), we would like to invite the resubmission of a significantly-revised version that takes into account the reviewers' comments.

Based on the reviewers' comments, this manuscript is of potential interest to the field, but the manuscript can be organized better: The method descriptions have room for improvement; the results should be described clearer and interpreted better. Specifically, none of the reviewers seems to be convinced that transcriptome diversity can be purely interpreted as a systematic bias. Substantial improvement on the data interpretation is warranted.

We cannot make any decision about publication until we have seen the revised manuscript and your response to the reviewers' comments. Your revised manuscript is also likely to be sent to reviewers for further evaluation.

Sincerely,

Chongzhi Zang

Guest Editor

PLOS Computational Biology

Jian Ma

Deputy Editor

PLOS Computational Biology

Based on the reviewers' comments, this manuscript is of potential interest to the field, but the manuscript can be organized better: The method descriptions have room for improvement; the results should be described clearer and interpreted better. Specifically, none of the reviewers is convinced that transcriptome diversity can be purely interpreted as a systematic bias. Substantial improvement on the data interpretation is warranted.

Reviewer's Responses to Questions

**Comments to the Authors:**

Reviewer #1: This paper introduces the concept of transcriptome diversity and defines it as the Shannon entropy of gene expression values within a given sample. The authors argue for importance of the concept from a few different aspects. The main take-away message is that PEER factors and PCs can be largely explained by transcriptome diversity, which authors believed to be largely correlated with confounders. Although as I understand it this message would not change current practices, this message may be informative to computational researchers.

I find the paper well-written overall. I provide some detailed suggestions below. I hope the authors can organize their work better and make the paper more valuable to the field.

Major points:

1. Specific subsections within MATERIALS AND METHODS should be referenced in main text and figure captions when relevant. For example, the paragraph starting in Line 201 uses the method described in Line 579 (I think), but it wasn’t specifically referenced. There are many instances like this throughout the results section.

2. In the current manuscript, sometimes the analyses are directly based on TPM or TMM-based expression values, other times the values are rankit-normalized before the analyses. Can these two approaches be unified to simplify the methodology? At the very least, it should be made much clearer in RESULTS and the figure captions which approach is used in each case, and the reasoning should be provided.

3. As shown in Fig. 2c, 96% of the TPM significant correlations are positive. Could the authors provide some explanations for this phenomenon?

4. For Figure 4, ordering the tissue types by sample size would be informative. Currently the tissue types are ordered by absolute value of Spearman correlation, it seems like. Ordering them by sample size may reveal additional pattern.

5. Is there any difference in transcriptome diversity for technical replicates?

6. For this manuscript, the authors treat the transcriptome diversity as a systematic source of bias in RNA-seq data. However, the difference of transcriptome diversity across samples itself may have some biological meaning. The authors should discuss this possibility.

7. As stated by the authors, TPM values have been shown to not properly account for sequencing depth differences across samples and for the influence of highly expressed genes on the rest of genes and are not proper for cross-sample comparison. Thus, the results from TMM-based expression values should be given more emphasis in the manuscript.

Minor points:

1. The authors should clarify how they calculate the gene expression variance in Fig. S1 and make use of “variance” or “variation” consistently between the x-axis label and the figure title.

2. For Fig. 6a-c, the gray-black color scale is not distinguishable (the scale is too subtle).

3. The points and violin plots in some figures of Fig. S7 and S8 overlap. Please adjust the width or transparency of the points.

4. Figure 2b and 2c: expression and transcriptome diversity are rankit-normalized? And r in the caption stands for Pearson correlation? Details like this should be made self-evident in the captions.

5. For MATERIALS AND METHODS section:

a. Line 469, Transcriptome diversity calculation (Shannon entropy)

i. It might be helpful to explain briefly why H ranges from 0 to log2(G) for readers who are not experts in information theory.

ii. I think it would be more straightforward to define p_i directly in terms of TPM, i.e., using the third equation on the first page of Additional File 4: Note S1, since a TPM value represents the relative abundance of a gene in a sample, the read lengths of genes already accounted for. More generally, p_i should be defined in terms of the normalized gene expression values, which would encompass TMM normalized values as well.

b. Line 549, PCA and clustering analysis:

i. The authors did the PCA analysis without scaling and centering. Are there any specific considerations for not using scaling and centering? In general, centering is almost always mandatory, and scaling is almost always preferred.

c. Line 570, Gene expression variance explained by PEER accounted by transcriptome diversity

i. In Line 575, the sentence “The difference of r2 values from expression associations between the intact PEER vs the “regressed out” PEER factors divided by the variance explained by PEER represents the amount of variance explained by PEER that can be accounted for by transcriptome diversity” needs to be rewritten. The word “between” does not make sense. And does “variance explained by PEER” refer to the R2 value between a gene expression vector and the intact PEER factors? This section may be better if the authors used mathematical notations and equations.

ii. This section seems to underly method behind Figure 5b. This section should be expanded to explain the methodology behind Figure 5b more comprehensively.

d. Line 579, Variance explained of gene expression matrices:

i. The description in this subsection would be made clearer if mathematical notations and equations are used.

Reviewer #2: The authors proposed a simple metric based on Shannon entropy to study the diversity of expression across genes within each sample. Through analyses, they found that the diversity measure is correlated with several biological variables and claimed it is a systematic source of bias. Below are some comments.

While it appears to be empirically correlated, I found it is hard to interpret the so-called bias. How does transcriptome diversity necessarily cause bias?

Fig 4a, the brain tissues are among the top ones with the highest absolute Spearman’s correlation between PC (principal component) and transcriptome diversity. Is there any biological reason for this? The clustering of brain tissues may be a sign that “transcriptome diversity” may not be pure bias, and there may be some underlying biological meaning.

The linear models on lines 526 and 542 should include an error term.

The description of the datasets appears twice in “Data Sources and data retrieval” and “Availability of data and materials.”

Reviewer #3: I think the authors have made important discoveries, however these discoveries can be interpreted better.

Here are suggestions to strengthen the manuscript:

1. The paper has been comparing TPM with TMM, which I do not think it is necessary. TPM is calculated by double normalization of gene length and library size. Essentially, it is a proportion estimate but not a quantity estimate. The metric can be used for within-sample comparison or clustering analysis. It is not a legitimate metric for cross-sample quantitative analysis. If a study has used TPM for eQTL analysis, that should consider as a common mistake. It is not appropriate to compare and benchmark with wrong metrics in a research paper, given that you already know its drawbacks. I suggest to include values normalized by median of ratios for comparison instead, as TMM takes into account length but median of ratios doesn’t.

2. It is not surprising that the transcriptome diversity is associated with PEER factors from bulk RNA-seq. In bulk RNA-seq, the transcripts are amplified during sequencing process, where they compete for chance for amplification. GC content, gene length and abundance are all factors that can determine how many of them get the slots during this stochastic process. But the primary contributor will be their relative abundances. This is also the reason that read count of a gene depends on all other genes. The degrees of freedom constrained by limited sequencing depth will naturally lead to our observed association between the transcriptome diversity and read counts. I suspect such association will not be as strong in data with Unique Molecule Identifiers (UMIs), such as some of single cell RNA-seq data, where the amplification bias has been mitigated by these identifiers.

3. The transcriptome diversity is manifestation of different regulation and functions of cell types. Besides technical factors, it should be confounded with biological factors such as cell types, developmental stages, transcriptional phase, genders, to name a few. That is to say, we cannot correct for the transcriptome diversity, otherwise we remove important biological variation. Although it can explain large portion of PEER factors, the metric itself does not dissect whether that portion is biological or technical. I suggest to regress PEER factors to biological factors and technical factors (sequencing depth in particular) to evaluate their contributions.

**Have the authors made all data and (if applicable) computational code underlying the findings in their manuscript fully available?**

Reviewer #1: None

Reviewer #2: None

Reviewer #3: Yes

PLOS authors have the option to publish the peer review history of their article (what does this mean?). If published, this will include your full peer review and any attached files.

Reviewer #1: No

Reviewer #2: No

Reviewer #3: **Yes: **Mengjie Chen
---

## [Decision Letter · Decision Letter 1]

1 Feb 2022

Dear Dr. Fraser,

Thank you very much for submitting your manuscript "Transcriptome diversity is a systematic source of variation in RNA-sequencing data" for consideration at PLOS Computational Biology. As with all papers reviewed by the journal, your manuscript was reviewed by members of the editorial board and by several independent reviewers. The reviewers appreciated the attention to an important topic. Based on the reviews, we are likely to accept this manuscript for publication, providing that you modify the manuscript according to the review recommendations.

As you can see, Reviewer #1 raised several additional comments which should be addressed to improve the accuracy and clarity of the manuscript.

Sincerely,

Chongzhi Zang

Guest Editor

PLOS Computational Biology

Jian Ma

Deputy Editor

PLOS Computational Biology

[LINK]

Reviewer's Responses to Questions

**Comments to the Authors:**

Reviewer #1: Line 10, “it is still challenging to detect and remove artifactual signals”: this statement may be inaccurate given the existence of methods such as PEER.

Line 30, “This is roughly similar to a doctor that treats only symptoms of a disease, since she does not know the underlying cause. Understanding the cause is not enough to cure the disease, but is a critical first step”: This analogy should be removed, in my opinion.

Line 91, “PEER factors a priori are not associated with any known biological or technical source”: they are associated, just not super strongly.

Line 115, “transcriptome diversity as measured by Shannon entropy [14] (Fig. 1; see Methods) across RNA-seq samples strongly correlates with the expression of many genes (measured by TMM)”: your Shannon entropy is calculated within each sample, across genes (not across samples).

Line 128, “We observed that most TMM-based expression values were correlated with transcriptome diversity (68.9% of genes at Benjamini-Hochberg False Discovery Rate (BHFDR) <0.05; see Methods). Interestingly, most (64%) of the significant correlations are positive”: it would be helpful to somehow indicate the significant correlations in the plot.

Line 170, “To our knowledge this is the first example of a known source of variability explaining PEER covariates to such an extent”: in GTEx data, how much variance do the known covariates explain? That info could be included here for comparison.

Response to previous comments, “We now define p_i in terms of TPM in the main text. We do not think it would be ideal to define p_i using TMM since this across-sample normalization would make the transcriptome diversity values depend on what other samples are being included and therefore more difficult to interpret”: this does not make sense. If the authors believe entropy shouldn’t be calculated based on TMM values, then how do the authors justify calculating entropy based on TMM values throughout the paper?

Response to previous comments, “We did try reordering this figure by sample size (shown below), but we didn’t see any clear pattern”: the tissue types should still be ordered by sample size to demonstrate there is no pattern, in my opinion.

Reviewer #2: My comments have been addressed.

Reviewer #3: The revision is satisfactory.

**Have the authors made all data and (if applicable) computational code underlying the findings in their manuscript fully available?**

Reviewer #1: None

Reviewer #2: None

Reviewer #3: Yes

PLOS authors have the option to publish the peer review history of their article (what does this mean?). If published, this will include your full peer review and any attached files.

Reviewer #1: No

Reviewer #2: No

Reviewer #3: No

Figure Files:

Data Requirements:

Reproducibility:

References:

---

## [Decision Letter · Decision Letter 2]

14 Feb 2022

Dear Dr. Fraser,

Thank you very much for submitting your manuscript "Transcriptome diversity is a systematic source of variation in RNA-sequencing data" for consideration at PLOS Computational Biology. As with all papers reviewed by the journal, your manuscript was reviewed by members of the editorial board and by several independent reviewers. The reviewers appreciated the attention to an important topic. Based on the reviews, we are likely to accept this manuscript for publication, providing that you modify the manuscript according to the review recommendations.

Specifically, please consider the reviewer's suggestion and further modify Figure 2. Once we receive a revised version, we can accept this manuscript for publication without sending to reviewers again.

Sincerely,

Chongzhi Zang

Guest Editor

PLOS Computational Biology

Jian Ma

Deputy Editor

PLOS Computational Biology

[LINK]

Please consider the reviewer's suggestion and further modify Figure 2. Once we receive a revised version, we can accept this manuscript for publication without sending to reviewers again.

Reviewer's Responses to Questions

**Comments to the Authors:**

Reviewer #1: "We have modified Fig 2 by adding a plot as Fig 2b to indicate the significant correlation in the plot." - it'd be great if the authors can just add two vertical lines indicating the thresholds beyond which the rho is considered significant.

**Have the authors made all data and (if applicable) computational code underlying the findings in their manuscript fully available?**

Reviewer #1: None

PLOS authors have the option to publish the peer review history of their article (what does this mean?). If published, this will include your full peer review and any attached files.

Reviewer #1: No

Figure Files:

Data Requirements:

Reproducibility:

References:

---

## [Editor Report · Decision Letter 3]

18 Feb 2022

Dear Dr. Fraser,

We are pleased to inform you that your manuscript 'Transcriptome diversity is a systematic source of variation in RNA-sequencing data' has been provisionally accepted for publication in PLOS Computational Biology.

Best regards,

Chongzhi Zang

Guest Editor

PLOS Computational Biology

Jian Ma

Deputy Editor

PLOS Computational Biology

---

## [Editor Report · Acceptance letter]

18 Mar 2022

PCOMPBIOL-D-21-01718R3 

Transcriptome diversity is a systematic source of variation in RNA-sequencing data

Dear Dr Fraser,

I am pleased to inform you that your manuscript has been formally accepted for publication in PLOS Computational Biology. Your manuscript is now with our production department and you will be notified of the publication date in due course.

With kind regards,

Livia Horvath
